# Body Dissatisfaction in Adolescents: Differences by Sex, BMI and Type and Organisation of Physical Activity

**DOI:** 10.3390/ijerph16173109

**Published:** 2019-08-27

**Authors:** Juan Gregorio Fernández-Bustos, Álvaro Infantes-Paniagua, Irene Gonzalez-Martí, Onofre Ricardo Contreras-Jordán

**Affiliations:** Faculty of Education, Albacete, University of Castilla La Mancha, 02071 Albacete, Spain

**Keywords:** body image, physical activity, adolescence, body satisfaction

## Abstract

The aim of this study was to assess the differences in body dissatisfaction (BD) of male and female adolescents by body max index (BMI) and the quantity, type and organisation of physical activity (PA). To do so, 652 adolescents aged 12–17 years participated in a cross-sectional study. The cognitive-affective component of BD was assessed with the Body Shape Questionnaire (BSQ) and the perceptual component with Gardner’s scale for the assessment of, body image (BI). PA was measured with the International Physical Activity Questionnaire (IPAQ-SF) and the item 1 from the Physical Activity Questionnaire for Adolescents (PAQ-A). The results show that sex and BMI are key variables when determining BD. Moderate-to-vigorous PA (MVPA) was moderately associated with a greater body satisfaction in males but no association was found between BD and the participation and organisation of PA. Moreover, the results suggest that participants in aesthetic/lean PA are at a higher risk of suffering from BD than participants in other PA types. These findings provide useful information for the design of programmes promoting healthy lifestyles, weight control and BI concern during the school period.

## 1. Introduction

Nowadays, body image (BI) is considered to be a construct composed of different dimensions. The perceptual dimension refers to how one sees oneself and describes one’s body, while the affective dimension focuses on feelings and emotions that one feels about one’s appearance. The cognitive dimension consists of the assessment of thoughts about body appearance and the behavioural dimension evaluates behaviours resulting from perceptions, thoughts and feelings about body appearance and function [1].

In recent decades, BI has received significant attention from researchers, since it is considered one of the most influential factors affecting psychological wellbeing. Along this line, body dissatisfaction (BD) is associated with anxiety, depression [2], eating disorders [3] and muscular dysmorphia [4]. The relationship between BD and adverse mental health could be due to shame and social isolation [5] and hormonal changes such as leptin [6].

Similarly, BI is a key element of the self-concept configuration, especially in adolescence [7,8]. We refer adolescence as “the period of life beginning with the appearance of secondary sex characteristics and terminating with the achievement of full maturity (usually 12–18 years of age)” [9] (p. 310), when everything related to the body becomes especially relevant. Older age, male gender and lower body mass index (BMI) were associated with better body esteem in adolescents [10]. Besides the mental health-related aspects, BD is prospectively related to unhealthy weight control behaviours, binge eating and lower levels of physical activity (PA) according to a 5-years longitudinal study [11]. On the other hand, high levels of body satisfaction can have protective effects, such as better eating habits, healthier levels of PA and a lower probability of suffering from overweight in late adolescence and early young adult ages [11,12].

In spite of this relationship between BI and health, BD reaches high levels during adolescence, especially among females [13] and other vulnerable groups, such as adolescents healed from paediatric leukaemia [14]. Some of the main reasons for this are the body changes taking place during this stage and the internalisation of the society-imposed aesthetic model [15]. This fact has led to BI research being more focused on females and more limited among males. Researchers have sometimes operationalised BD in relation to the desire to be thinner, which results in a too simplistic view for male BI ideals. Opposite to the thin ideal for females, men from occidental societies are led to desire both getting away from obesity and being more muscled [4]. As both overweight and underweight can be risk factors for BD, the initial research supposing a linear association between BMI and dissatisfaction in females [16] may change to be curvilinear in males [17], since the latter can suffer dissatisfaction from underweight and overweight [18]. In other words, associations between BD and BMI continue to be gender-specific [19].

Researchers have also been concerned about the relationship between BI and PA. The benefits of PA to health—including mental health—have been proved to the extent that a set of international recommendations [20] have been established on the quantity and type of PA to ensure those benefits. Nevertheless, the relationship between PA and BI is complex, as seen from the uneven results of studies. Many studies focused on proving the positive associations of PA and sports with a more positive BI [21] or even on how PA-based interventions can improve BI [22]. Previous reviews and meta-analyses have echoed the positive relationship between BI and PA in males and females but the moderating role of sex is unclear. Some authors found a higher effect among females [23] but others did not [24,25]. Among the limited specific literature on males, Bassett-Gunter, McEwan and Kamarhie [26] showed how better participation in PA was associated with positive BI. However, among female adolescents, McIntosh-Dalmedo et al. [27] did not find enough evidence to suggest that sport and exercise interventions can improve BI. These heterogeneous results highlight the possible existence of differentiating mechanisms for males and females when relating PA and BI.

Besides gender, the literature has made the effort to find other moderators in the relationship between PA and BI, such as adiposity or the intensity, quantity and type of PA. Regarding adiposity, a positive correlation between BD and BMI in a general population [28] as well as in an athletic population was observed. For example, Karr et al. [29] found that a higher BMI was associated with a higher BD in female participants of different types of sports.

Some studies concluded that vigorous PA had a higher impact on BI when compared with light PA, which had practically no effect [23,30], while others did not reach this outcome and only found a relationship with light PA in males [26]. With respect to quantity of PA, some researches showed that higher levels of PA are associated with a protective effect on BD, regardless of BMI and sex [13,31]. Nevertheless, more research is required since the number of studies on which these conclusions are based is limited.

Concerning the moderating effect of the type of PA on dissatisfaction, results were also inconclusive, although it must be considered that different criteria for PA classification have been used. On the one hand, even though Hausenblas and Fallon [23] supported the theory that a combination of aerobic and anaerobic PA had a higher effect on BD, a meta-analysis showed a higher effect of anaerobic activities [30] and other researchers concluded that the type of PA did not generally moderate the relationship between PA and BI [26,32]. When comparing endurance-based with strength-based interventions in females, Martin-Ginis et al. [22] showed that endurance workouts produced better effects on BI than strength workouts. Moreover, the literature persevered in distinguishing among aesthetic/lean sports (e.g., gymnastics), non-aesthetic/lean sports (e.g., athletics) and non-aesthetic/non-lean sports (e.g., volleyball). Although there is more evidence suggesting that participation in aesthetic/lean sports and, to a lesser extent, in non-aesthetic/lean sports is associated with higher BD than participation in a non-aesthetic/non-lean sport [33,34,35,36], results are contradictory. Thus, some studies in female adults [37] and adolescents [38] did not find differences on satisfaction according to the type of sport—aesthetic vs. non-aesthetic—and only BMI had a significant influence on dissatisfaction [29]. In a meta-analysis based on 10 studies concerning athletic females, Varnes et al. [39] found that participants in sports that emphasised physical appearance (e.g., gymnastics) showed higher BD than participants in non-aesthetic/non-lean sports (e.g., basketball). Some studies have supported this view. For example, Robbeson, Kruger and Wright [40] found higher BD among dancers, while Russel [41] observed that non-aesthetic sport players such as basketball, cricket or rugby players had more positive perceptions of their BI. In contrast, De Bruin, Oudejans and Bakker [42] found the same perception of body shape and size between female gymnasts and non-gymnasts. Similarly, in a recent study on female top-level athletes, those who practiced aesthetic sports showed a more positive BI than non-aesthetic sports participants [43].

Studying the interaction between gender and type of sport could have interesting results. Nevertheless, only a few studies have compared results between males and females in the same research or have been uniquely focused on males. Perelman et al. [44] found a higher BD among female college students compared to males, regardless of the type of sport. Additionally, these authors suggested that athletes participating in sports that promote lean body types were at a higher risk of developing significant BD. Other studies demonstrated that track athletes had higher BD than martial arts practitioners, football players or non-athletes in males [45] and females [46].

While there seems to be differences between lean sports and non-lean sports, discrepancies in the literature make clear that more research on BD and type of sport is needed. On the other hand, in addition to the disproportionately high number of research studies on females in comparison to research including males, most of these studies assessed only one aspect of dissatisfaction which is not enough for males: the desire to lose weight. For the aforementioned reasons, it is necessary to gain a better understanding of the relationship between PA and the different dimensions of BI in adolescents, by considering the possible differences between male and female adolescents and including other aspects of BD in males (i.e., desires to gain weight). From a research perspective, not controlling for moderators in the relationship between PA and BI may result in a moderate or null effect if exercise benefits vary according to the participants’ characteristics (BMI, gender) or PA characteristics (quantity and type) [47]. Therefore, the aim of this study was to assess the differences in BD of male and female adolescents by BMI, quantity and type of PA. Additionally, this study also included the organisation of PA as a new moderating variable. To do so, BI was assessed by means of cognitive-affective components as well as the perceptual component.

## 2. Materials and Methods

### 2.1. Participants

This research was carried out on a sample of 652 adolescents (296 male and 356 female) aged 12–17 years (M = 14.57, SD = 1.51) of La Roda (Spain). The students attended the courses corresponding to the compulsory secondary education of the Spanish education system, which coincide with ages comprised in adolescence (12–17 years). This number includes those students who were available on the day of data gathering and whose parents had authorised them to participate after being informed of the study. The final sample, however, was composed of 634 participants (284 male and 350 female), since data from 18 subjects were deleted due to errors in data filing and input. The response rate was approximately 80%.

### 2.2. Measures

In order to assess the cognitive-affective components as well as the behavioural component of BI, a cross-cultural adapted and validated to the Spanish context version [48] of the Body Shape Questionnaire (BSQ) [49] was used. This questionnaire measures the dissatisfaction provoked by the body itself, the fear of gaining weight, physical appearance self-devaluation, the desire to lose weight and the avoidance of situations in which one’s physical appearance could gain attention from others (e.g., “Have you been so worried about your shape that you have been feeling you ought to diet?”). The questionnaire consists of 34 self-administered items on a six-point Likert scale (i.e., 1 = never, 2 = rarely, 3 = sometimes, 4 = often, 5 = very often, 6 = always). When the total sum of the items’ scores exceeds 81 points it is considered that a concern for BI exists. The BSQ showed an internal consistency of α = 0.97 for this research.

The perceptual component was assessed using Gardner’s scale for the assessment of BI [50], which was adapted to the Spanish population by Rodríguez et al. [51]. It is composed by 13 silhouettes representing a schematic contour of the human shape with no additional characteristic such as hair, facial features and so forth. These silhouettes were developed by following data from the National Center for Health Statistics from the United States. The middle figure represents the median distribution of weight for the reference population. This middle silhouette was modified by increasing or decreasing 30% of its volume to create six different silhouettes at each of its sides. These silhouettes represent a growing order of weight increase of 5% for each silhouette towards the right and a decreasing order of 5% weight reduction for each silhouette towards the left. Therefore, a continuum of silhouettes, whose extremes represent an extremely thin or obese figure, is displayed. This scale allows for a self-assessment of the perception of an individual’s body as well as the figure that represents one’s ideal to reach; difference between both silhouettes points out the discrepancy between the desired and the perceived body shape (DDPB). The greater the discrepancy, the higher the BD. The central figure is assigned a weight of 0, whereas figures on its left are assigned negative values (from −1 to −6) and figures on its right are assigned positive values (from 1 to 6). The DDPB value is calculated by subtracting the value of the perceived body shape from the value of the desired body shape. Positive values thus represent the desire to gain weight and negative values, the desire to lose weight.

Following International Society for the Advancement of Kinanthropometry (ISAK) standardised guidelines, participants were individually weighted and measured by two Level 1 ISAK researchers. To do so, a calibrated digital scale Tanita, model UM-075 (with a sensibility of 0.1 kg), and a 2 m altimeter Holtex (Holtex, Aix-en-Provence, France) were used. Students were measured and weighted wearing light clothes (no long trousers, sweaters or shoes were allowed). Two measures of body weight and height were taken and their averages were noted. After calculating BMI scores, participants were classified according to the eight cut-offs proposed by the International Obesity Task Force (IOFT) [52] by age and sex. These cut-off scores were operationalised into four categories: underweight, normal weight, overweight and obesity.

To estimate the adolescents’ levels of PA, the short form of the International Physical Activity Questionnaire (IPAQ-SF) was used. This is a self-reported questionnaire whose reliability and validity have been tested within different contexts [53], including in adolescents from Spain [54]. This study relies only on the data on frequency and duration of both moderate and vigorous PA. From these data, participants were classified into inactive, active not meeting the recommendations of moderate-to-vigorous PA (MVPA) and active meeting the recommendations of MVPA. Weekly MVPA was also calculated.

In order to know the type of PA practiced, participants also answered item 1 (“Physical activity in your spare time: Have you done any of the following activities in the past 7 days (last week)? If yes, how many times?”) from the Spanish version [55] of the Physical Activity Questionnaire for Adolescents (PAQ-A) [56]. Participants were also advised that the sport participation did not include the sports practised during their physical education lessons. When participating in different types of PA, the most frequently practised type was only considered.

Once these data were collected, sport participation and PA were classified into three types according to the categorisation system followed by previous studies [29,57,58]: non-aesthetic/non-lean—activities or sports in which the performance does not depend on appearance or slimness (e.g., basketball, football, tennis); non-aesthetic/lean—the performance is not based on appearance but on a lean physical body (e.g., cycling, swimming, athletics); and aesthetic/lean—activities whose purpose is to improve the physical appearance or for which physical appearance is important to success (e.g., fitness, gymnastics, aerobics).

Additionally, a question on the organised or non-organised nature of PA was included. That is, whether PA was planned and implemented under an instructor or trainer’s supervision or, in contrast, there was no external guidance.

### 2.3. Procedure

A cross-sectional design was followed for the present research. Participants were recruited from the entire population of secondary education students in La Roda (Spain). Before data gathering, educational boards and parents of the participants were informed about the aims, procedures, risks and benefits of the study. School boards were initially contacted via telephone and, subsequently, an informative meeting was set. The participants’ parents were sent a letter from the schools through the supervising teachers of each class. All students with the parents’ consent participated voluntarily. This study conformed to the deontological guidelines defined by the Declaration of Helsinki (Hong Kong revision, 1989) and followed the recommendations of the Good Clinical Practice in the European Community (1 July 1991, document 111/3976/88) and the Spanish legislation on human clinical research (Royal Decree 561/1993 on clinical trials). Therefore, the anonymity of the participants and data confidentiality were guaranteed. The protocol was approved by the Ethics Committee on Human Research (University of Castilla La Mancha; reference: 201905054).

Questionnaires were administered to groups of 20 to 30 students in a wide room in order to ensure participants’ focus and privacy. Once the questionnaires were distributed, a researcher verbally explained the guideline to correctly complete them, remarking the importance of being careful and honest when answering. These guidelines were also given in writing before completing the questionnaires. Participants were informed about the anonymity of the questionnaires in order to avoid social desirability in some answers. The participants were given 25 min to fulfil all the questionnaires, which was verified to be enough time to answer every proposed item.

Finally, students were weighted and measured by two expert researchers in a closed room, which also guaranteed the privacy and confidentiality of data gathering.

### 2.4. Data Analysis

All data were analysed using SSPS Statistics 24 (IBM Corporation, Armonk, NY, USA) software. Statistical significance was set at *p* = 0.05. Tests of normality (Kolmogorov-Smirnov) and homoscedasticity (Levene) were run. The assumptions were not met for all the variables; however, due to the high number of participants included into our sample, we opted to use parametric statistical tests by assuming the central limit theorem. Descriptive analyses (means and standard deviations) were run for dependent variables (e.g., scores of BSQ and DDPB); male and female participants were also compared through t tests. Pearson correlation coefficient (r) was utilised to examine the relationship between BSQ, DDPB, BMI and MVPA (minutes per week). A chi-square test (χ^2^) examined differences between males and females in frequencies by category of BMI and organisation, type and level of practice of PA. A factorial ANOVA was run to explore the main effects of independent variables (sex, BMI and type and practice of PA) and their interactions on BSQ and DDPB scores. Among the participants who were categorised as active, a MANCOVA with BMI as a covariate was run to determine the effect of sex and type and organisation of PA on BD (score of BSQ and DDPB). Since the factors and covariate included in the MANCOVA were significant (Pillai’s Trace *p* < 0.05), two different ANCOVAs were run to see the effects of the included factors on BSQ and DDPB separately. Pairwise comparisons (Bonferroni correction) controlling for BMI were also conducted in order to determine the differences by type of PA on BSQ and DDPB. Eta squared and partial eta squared were computed as sample size measures.

## 3. Results

There were no differences for mean age and BMI between males and females (*p* > 0.05). The males’ amount of weekly PA was three times the females’ weekly PA (♂ M = 281.46, SD = 203.19; ♀ M = 90.45, SD = 137.10). Females, however, showed notably higher concern about their bodies than males (*p* < 0.001). In fact, the female group’s mean score on the BSQ (M = 85.14, SD = 39.99) exceeded the established cut-off score for BI concern (>81), whereas among males that score was not reached even in the obese male group (M = 76.56, SD = 33.56). Except for the underweight group (M = 1.39, SD = 1.24), females reported a greater DDPB (*p* < 0.001)—they desired to be thinner than what they perceived themselves to be. Underweight (M = 2.44, SD = 2.12) and normal weight males (M = 0.30, SD = 1.70) desired a bigger body than the perceived. Additionally, organised PA was more frequent than non-organised PA, especially among females (organised 81.2%, non-organised 18.8%). Non-aesthetic/non-lean PA was the most practised PA type in males (59.4%), while aesthetic/lean PA was the most frequent for females (46.2%). Within females, the percentage of inactive participants was high (62.9%) and the group meeting the daily MVPA recommendations was low (9.1%). All measures’ values are reported in Table 1 and Table 2.

Correlations between BD, BMI and weekly MVPA are reported in Table 3. BD measured by the BSQ (cognitive-affective-behavioural component) was significantly correlated (*r* = −0.697; *p* < 0.01) with more DDPB (perceptual component). Both types of dissatisfaction were associated with a higher BMI, especially in females’ discrepancies (*r* = −0.676; *p* < 0.01). Weekly MVPA was only correlated with a lower dissatisfaction (BSQ *r =* −0.103, DDPB *r =* 0.155; *p <* 0.01) and a lower BMI (*r =* −0.196; *p <* 0.01) in the case of males.

ANOVA results are shown in Table 4. These data highlight the significant effect of sex and BMI on the variance of the BSQ score (*p* < 0.001) and DDPB (sex *p <* 0.001; BMI *p* = 0.003), as well as the interaction between both on the BSQ (*p* = 0.025). However, there was no difference in BD (*p* > 0.05) among the different established groups based on the PA practice (inactive, active not meeting the MVPA recommendations and active meeting the MVPA recommendations). The interactions of PA practice with the rest of the independent variables were not significant in BD.

In order to determine the different effects of PA types and their organisation on BD among the active participants, a MANCOVA was run. This way the associations between the two measures of BD were considered, while BMI was also included as a covariate to control its effect. The results, reported in Table 5, show significant differences (*p* ≤ 0.001) in the dissatisfaction according to PA type (i.e., non-aesthetic/non lean, non-aesthetic/lean or aesthetic/lean), even after interaction with sex (BSQ *p* = 0.38, DDPB *p* = 0.009). However, PA organisation did not influence the different types of dissatisfaction (*p >* 0.05). Only the interaction between PA organisation and type had a significant effect on DDPB, where participants in organised non-aesthetic/non-lean PA showed a greater discrepancy.

Eventually, a pairwise analysis was carried out to establish the concrete differences between the groups according to the type of PA. The inactive group was also included in the analysis. Table 6 shows that in all cases, for the same type of PA, males reported less dissatisfaction than females. In the case of the BSQ score, both male and female participants in aesthetic/lean activities reported higher levels of dissatisfaction (♂ M = 67.00, SD = 36.74; ♀ M = 95.95, SD = 42.68). Among females, inactive subjects also showed higher levels of dissatisfaction (M = 87.50, SD = 39.79) than non-aesthetic PA participants (non-lean: M = 69.06, SD = 33.77; lean: M = 67.22 SD = 32.80). For the DDPB, significant differences were only found for females, with the inactive (M = −2.37, SD = 2.72) and aesthetic PA participants (M = −2.60, SD = 2.35) showing a higher discrepancy in terms of the desire to be thinner when compared to non-aesthetic PA participants (non-lean: M = − 1.53, SD = 2.44; lean: M = − 1.04, SD = 1.91).

## 4. Discussion

This research aimed to determine the effect of variables such as sex, BMI and the level, type and organisation of PA on adolescents’ BD. The results support the outcomes of previous research, highlighting sex and BMI as crucial variables when determining dissatisfaction in adolescents [16,19]. Female adolescents indeed showed higher concerns about their BI [13] in terms of both the desire to lose weight and a higher DDBP, regardless of BMI. This fact can be explained by the greater social pressure that female adolescents are subjected to, together with the early internalisation of the aesthetic model—which identifies a thin body as key to being attractive. This conflicts with the body changes suffered during this period in life [15]. On the part of adiposity, a higher BMI was associated with higher BD [28] and depression in adolescent females [59]. Nevertheless, contrary to females, dissatisfaction levels in males were low, even among obese participants. Regarding the DDBP, overweight and obese males desired thinner bodies, while underweight and normal weight males showed a positive discrepancy—that is, they desired to be bigger than they perceived themselves to be. Obese and underweight males were indeed the most dissatisfied with their physical appearance at the perceptual level [16]. This circumstance is compatible with the social pressure suffered by males, not only to distance oneself from obesity but also to be bigger and more muscled [60]. They therefore suffer from dissatisfaction with underweight and overweight [18]. To sum up, it is highlighted that relationships between BD and BMI are gender-specific [19].

On the other hand, the present study also supports the idea of different extant mechanisms regulating the relationship between PA and BI by gender [61]. First, a positive but small correlation between MVPA and body satisfaction, as well as a negative correlation between MVPA and BMI, were found in males. In females, however, none of these relationships were confirmed. These findings are partially compatible with those of other authors,’ confirming that a higher level of PA has a protective effect against dissatisfaction [13,31]. Additionally, our results confirm the evidence found by Altintas et al. [62] in both male and female adolescents. Bassett-Gunter et al. [26] and McIntosh-Dalmedo et al. [27] also found positive correlations between PA and BI in male and not in female adolescents, respectively. However, our results also contradict other studies with adult samples, which suggested that PA is related to BI improvements in both males and females [23,24,25].

Despite the relationship found between MVPA levels and lower dissatisfaction in males, there was no evidence confirming that PA practitioners (i.e., active subjects) were more satisfied with their bodies than inactive subjects. An important part of the literature establishes that athletes have a more positive BI compared with non-athletes [37]. However, some studies reported a lack of association between PA participation and BI [63], even in adolescence [64,65]. Along this line, Sabiston et al. [25] advised that many null effects between both variables were reported when controlling for the effect of BMI, as also occurred in the present study.

To state that PA is or is not related to BI improvements is too unspecific since the characteristics, typology, organisation of and reasons for PA practise are diverse. For this reason, in this study we intended to further examine some moderating variables such as the type of PA or sport practised and its organisation by isolating the effect of BMI. In contrast to previous studies [37,38,42], our results show that PA type was a determining variable on the BI perceptions for both sexes, regardless of BMI. By analysing these results more exhaustively, males again showed lower levels of dissatisfaction, independent of the type of sport. This is in line with the findings of Perelman et al. [44]. Additionally, differences between male and female participants were found for the different types of sports according to the dependent study variables (BSQ and DDBP). In females, BI concerns were significantly lower among those who practised non-aesthetic/non-lean and non-aesthetic/lean sports as compared to those who practised aesthetic/lean PA or were inactive, in terms of the desire to lose weight and the discrepancy. In males, however, only aesthetic/lean PA participants were more concerned about their BI in the BSQ score. Apart from that, non-aesthetic/lean adolescent PA participants were the only group that showed a desire to be bigger. These results support the theory stating that aesthetic/lean sports participation is linked to higher BD in both females [33,34,39,40,41] and males [35,36,44].

Researchers have made efforts to explain why differences can exist in BD between different physical sport activities. The most traditional approach is focused on the pressures that the characteristics of each type of sport put on their participants, which somehow determines their success [66]. However, recent research notes that the sexual objectification of female athletes in the media may be a possible additional pressure. This leads females to define themselves by their body and appearance by internalising concrete beauty ideals that can increase BD in some athletes [39,67]. Although men are less likely to suffer from sexual objectification than women [68], the former are increasingly feeling more dissatisfied with their bodies while struggling to adapt to the ideal body for their sport [69]. Internalisation of the body ideal increases BD in both females and males, influenced by the pressure of maintaining a specific body shape and weight to optimally perform according to each sport’s demands. For this reason, researchers maintain that the greater sexual objectification of appearance-focused (i.e., aesthetic) sports participants comes with a higher risk of BD [29,30].

On the other hand, another important aspect emerges from that internalisation of socially imposed models for males and females—the reason for adolescents to practise some types of PA. The aesthetic reasons indeed mediate the relation between the internalisation of societal standards of attractiveness and BI [70]. That is, the dissatisfaction with one’s own body may itself lead some adolescents to take part in certain types of PA—which are supposedly oriented to physical attractiveness—with the aim of producing changes that will improve their appearance [71]. This is why practising PA for reasons such as weight loss, appearance improvement or increasing muscle tone is systematically associated with a more negative BI [70,72]. This is usually channelled through the practice of aesthetic/lean sports [71].

The greater concern of aesthetic/lean athletes may explain the lack of relationship between PA and BI, especially among females, since their rate of participation in this type of PA is far higher than others. Inactive subjects indeed showed similar levels of dissatisfaction to aesthetic sports participants. In the case of females, these levels were significantly lower when compared with non-aesthetic sports participants. That is, PA related to non-aesthetic (lean or non-lean) sports might have a protective effect against dissatisfaction. This would also corroborate the positive relationship between PA and BI shown in many previous studies [23,24,37]. Nevertheless, since the present results are based on cross-sectional data, these cannot provide evidence on whether the direction of the effect may go from the type of sport to concerns on BI or, contrary, higher concerns on BI may determine participation in aesthetic sports.

Finally, another mediating variable that has previously been addressed in the literature on the relationship between PA and BI is the sport level. Some studies determined that high-level athletes were more satisfied with their bodies [43], while others associated a greater competition level to higher BD [73]. Since most of the adolescent students that participated in our study did not take part in high-level sports, the moderating role of the organisation variable was studied instead. The aim was to determine whether individuals practising organised sports under a trainer’s supervision differed in terms of dissatisfaction from those whose sport practice was self-guided. Results showed that organisation did not affect the athletes’ BI, which means that BD does not rely on the organisation or level of competition but rather on the type of the PA practised [39,44].

Despite the revealed results, this study presented some limitations that must be considered. First, as previously stated, this is a cross-sectional study and therefore causal relationships cannot be established. In addition, more sensitive instruments to measure dissatisfaction with musculature in males would have provided a closer view on how they report BD related to PA. We must also highlight that this study was strictly focused on PA and sports practice; thus, important variables on sports participation, such as motivation, were put aside. These would have been of great value to explain some of the obtained results. Altogether, these results confirm the complexity of the study of PA and BI.

## 5. Conclusions

In summary, the results of this study emphasised gender and BMI as crucial variables when determining dissatisfaction in adolescents. Concretely, female participants reported greater desires to lose weight and a greater DDPB regardless of BMI. MVPA was moderately related to satisfaction in males, albeit no association was found between PA participation and better physical perceptions in both sexes. Aesthetic/lean activity participants showed higher levels of BD than non-aesthetic (lean or non-lean) activity participants. Inactive subjects, on their side, reported similar levels of dissatisfaction to aesthetic PA participants and higher levels than non-aesthetic PA participants. This could explain the lack of relationship between PA and BI found in previous research. Finally, PA organisation was not considered to be a moderating variable. In short, it is highlighted that not only the relationship between BD and BMI, but also their relationships with PA, are gender-specific. The results may have implications for the promotion of health in adolescents. These could be useful for designing school- and sport club-based programmes focused on healthy lifestyles, weight control and BI concerns.

## Figures and Tables

**Table 1 ijerph-16-03109-t001:** Descriptive statistics for the sample (*N* = 634).

Variables	Male (*n* = 284)		Female (*n* = 350)		All (*n* = 634)		*p*-Value	η²
	Mean	SD	Mean	SD	Mean	SD		
Age	14.50	1.51	14.66	1.50	14.59	1.51	0.177	0.00
BMI	22.36	3.93	21.81	3.64	22.05	3.78	0.069	0.00
PA minutes per week	281.46	203.19	90.45	137.10	176.01	194.57	0.000 ***	0.24
BSQ								
Underweight	38.78	8.98	48.56	13.23	45.30	12.70	0.000 ***	0.14
Normal weight	47.16	20.58	78.57	36.21	65.48	34.34	0.000 ***	0.21
Overweight	63.91	30.01	105.14	40.74	84.40	41.20	0.000 ***	0.25
Obesity	76.56	33.56	119.72	43.08	93.82	42.90	0.000 ***	0.25
Total	54.41	26.80	85.14	39.99	71.37	37.91	0.000 ***	0.16
DDPB								
Underweight	2.44	2.12	1.39	1.24	1.74	1.63	0.115	0.09
Normal weight	0.30	1.70	−1.67	2.32	−0.85	2.30	0.000 ***	0.18
Overweight	−2.01	1.85	−4.19	2.18	−3.09	2.29	0.000 ***	0.23
Obesity	−3.07	1.38	−4.22	1.43	−3.53	1.50	0.010 **	0.14
Total	−0.60	2.21	−2.21	2.61	−1.49	2.56	0.000 ***	0.10

** *p* < 0.01; *** *p* < 0.001. DDPB: Discrepancy Desired-Perceived Body. BMI: Body Mass Index categorised by IOTF (2012). PA: physical activity. BSQ: Body Shape Questionnaire. *p*-value from independent t-tests comparing male and female participants. η²: eta squared.

**Table 2 ijerph-16-03109-t002:** Frequency by category of BMI and organisation, type and practice of physical activity (PA).

Variable	Group	Males		Females		All		*x* ^2^
		*N*	%	*n*	%	*n*	%	
BMI	Underweight	9	3.2	18	5.1	27	4.3	9.17 *
	Normal weight	168	59.2	235	67.1	403	63.6	
	Overweight	80	28.2	79	22.6	159	25.1	
	Obesity	27	9.5	18	5.1	45	7.1	
PA Organisation	Organised	141	66.5	108	81.2	249	72.2	8.78 **
	Non-organised	71	33.5	25	18.8	96	27.8	
Type of PA	Non-aesthetic/non lean	129	59.4	47	36.2	176	27.8	44.30 ***
	Non-aesthetic/lean	58	26.7	23	17.7	81	12.8	
	Aesthetic/lean	30	13.8	60	46.2	90	14.2	
PA Practice	Inactive	67	23.6	220	62.9	287	45.3	146.31 ***
	<60 min/day of MVPA	76	26.7	98	28	174	27.4	
	≥60 min/day of MVPA	141	49.6	32	9.1	173	27.2	

* *p* < 0.05; ** *p* < 0.01; *** *p* < 0.001. MVPA: Moderate to Vigorous Physical Activity.

**Table 3 ijerph-16-03109-t003:** Correlations between body dissatisfaction, body mass index (BMI) and moderate–vigorous physical activity (MVPA).

Variable	DDPB	BMI	MVPA (min/week)
BSQ	−0.697 ** (♂ −0.538 **, ♀ −0.719 **)	0.318 ** (♂ 0.425 **, ♀ −0.423 **)	−0.268 ** (♂ −0.103 **, ♀ −0.090)
DDPB		−0.523 ** (♂ −0.676 **, ♀ −0.556 **)	0.249 ** (♂ 0.155 **, ♀ 0.087)
BMI			−0.025 (♂ −0.196 **, ♀ −0.007)

** *p* < 0.01. DDPB: Discrepancy Desired-Perceived Body. BMI (IOTF): Body Mass Index categorised by IOTF (2012). MVPA (min/week): Total minutes a week of Moderate to Vigorous Physical Activity.

**Table 4 ijerph-16-03109-t004:** Three-way Analysis of Variance for body shape questionnaire (BSQ) and Discrepancy Desired-Perceived Body.

Varaibles	*df*	BSQ		*ηp²*	DDPB		*ηp²*
		F	*p*		F	*p*	
Sex	1	25.82	0.000 ***	0.06	33.57	0.003 **	0.03
PA practice	2	0.01	0.983	0.00	1.17	0.728	0.00
BMI	3	14.09	0.000 ***	0.13	151.67	0.000 ***	0.27
Sex × PA practice	2	0.29	0.745	0.00	0.80	0.804	0.00
Sex × BMI	3	2.31	0.025 *	0.01	3.05	0.562	0.00
PA practice × BMI	3	1.09	0.363	0.01	1.61	0.956	0.00
Sex × PA practice × BMI	3	1.02	0.418	0.00	1.82	0.877	0.00
*R* ^2^		−0.36			−0.48		

* *p* < 0.05; ** *p* < 0.01; *** *p* < 0.001. DDPB: Discrepancy Desired-Perceived Body. *ηp*²: partial eta squared.

**Table 5 ijerph-16-03109-t005:** MANCOVA for BSQ and Discrepancy Desired-Perceived body and BMI as a covariate.

Variables	*df*	BSQ		*ηp²*	DDPB		*ηp²*
		F	*p*		F	*p*	
BMI	1	63.50	0.000 ***	0.14	203.32	0.000 ***	0.30
Sex	1	39.95	0.000 ***	0.10	31.62	0.000 ***	0.07
Type of PA	2	14.25	0.000 ***	0.09	6.28	0.001 **	0.04
Organisation	1	.125	0.724	0.00	0.023	0.879	0.00
Sex × Type of PA	2	3.31	0.038 *	0.02	4.28	0.009 **	0.03
Sex × Organisation	1	1.25	0.263	0.00	1.91	0.167	0.00
Type of PA × Organisation	2	0.85	0.425	0.00	4.00	0.019 *	0.02
Sex × Type of PA × Organisation	2	2.25	0.107	0.01	0.365	0.694	0.00
*R* ^2^		0.37			0.49		

* *p* < 0.05; ** *p* < 0.01; *** *p* < 0.001. DDPB: Discrepancy Desired-Perceived Body. *ηp*²: partial eta squared.

**Table 6 ijerph-16-03109-t006:** Means for BSQ and Discrepancy by type of PA. Pairwise comparisons (Bonferroni).

Variable	Group	Males			Females			*p*-Value	*ηp²*
		*n*	Mean	SD	*n*	Mean	SD		
BSQ	Non-aesthetic/non-lean (a)	129	52.75	26.93	47	69.06	33.77	0.001 **	0.08
	Non-aesthetic/lean (b)	58	45.26	14.52	23	67.22	32.80	0.000 ***	0.24
	Aesthetic/lean (c)	30	67.00	36.74	60	95.95	42.68	0.002 **	0.16
	Inactive (d)	67	59.88	26.76	220	87.50	39.79	0.000 ***	0.16
	*p*-value (Pairwise comparisons)	0.036 * (a, b < c)			0.000 *** (a < c, d; b < c)				
DDPB	Non-aesthetic/non-lean (a)	129	−0.61	1.95	47	−1.53	2.44	0.011 *	0.07
	Non-aesthetic/lean (b)	58	0.29	2.05	23	−1.04	1.91	0.009 **	0.14
	Aesthetic/lean (c)	30	−0.90	3.19	60	−2.60	2.35	0.005 **	0.23
	Inactive (d)	67	−1.22	2.08	219	−2.37	2.72	0.002 **	0.11
	*p*-value (Pairwise comparisons)	0.562			0.001 *** (a > c, d; b > c)				

* *p* < 0.05; ** *p <* 0.01; *** *p* < 0.001. DDPB: Discrepancy Desired-Perceived Body. *ηp*²: partial eta squared.

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
