# Peer review of "Body Dissatisfaction in Adolescents: Differences by Sex, BMI and Type and Organisation of Physical Activity"

_ijerph, 2019, doi:10.3390/ijerph16173109_

Round 1

Reviewer 1 Report

Thank you for inviting me to review the paper on “Body Dissatisfaction in Adolescents: Differences by Sex, BMI and Type and Organisation of Physical Activity”. This study has large sample size with adequate measures on body image and satisfaction. This is a good paper and deserves to be published in IJERPH. I have the following recommendations:

1.     Under introduction, line 33, the authors stated “Along this line, body dissatisfaction (BD) is associated with anxiety, depression [2], eating disorders [3] and muscular dysmorphia [4].” This is a simplistic view. I recommend the authors to expand this statement by mentioning the biological and psycho-social explanation.

Line 34… depression [2], eating disorders [3] and muscular dysmorphia [4]. The relationship between BD and adviser mental health could be due to shame and social isolation (Tran et al 2019) and hormonal changes such as leptin (Yang et al 2016).

References:

Tran BX et al Global Evolution of Obesity Research in Children and Youths: Setting Priorities for Interventions and Policies. Obes Facts. 2019;12(2):137-149.

Yang JL et al  The Effects of High-fat-diet Combined with Chronic Unpredictable Mild Stress on Depression-like Behavior and Leptin/LepRb in Male Rats. Sci Rep. 2016 Oct 14;6:35239. 

2.     Under Introduction line 36, the authors stated “Similarly, BI is a key element of the self-concept configuration, especially in adolescence [5,6], when everything related to the body becomes especially relevant.” I think the authors need to be more specific because the statement “when everything related to the body becomes especially relevant is rather non-specific.  Please add the following statement:

`     Line 37 …… becomes especially relevant. Older age, male gender and lower body mass index were associated with better body esteem in adolescents (Mak et al 2013).

Reference: 

Mak KK et al (2013) Body esteem in Chinese adolescents: effect of gender, age, and weight. J Health Psychol. 2013 Jan;18(1):46-54. PMID: 22373813

3.     Under discussion, line 292, the authors mentioned that “On the part of adiposity, a higher BMI was associated with higher BD [23]” I suggest the authors to indicate the relationship between high BMI and depression in female adolescents. Please revise the statement as follows:

Line 293: higher BMI was associated with higher BD [23] and depression in adolescent females (Quek et al 2017).

Reference

Quek YH et al Exploring the association between childhood and adolescent obesity and depression: a meta-analysis. Obes Rev. 2017 Jul;18(7):742-754.

 4.     Line 373, under limitations, the authors should state the limitation of cross-sectional study and cannot establish causality.

Author Response

We would like to thank the reviewer for providing us with such a positive assessment on our article, as well as for the comments reported to improve this work. In the following, we provide details on the enhanced aspects based on the reviewer’s suggestions. All recommended references have been included.

REVIEWER´S COMMENTS

AUTHOR´S RESPONSE

CHANGES IN TEXT

1.     Under introduction, line 33, the authors stated “Along this line, body dissatisfaction (BD) is associated with anxiety, depression [2], eating disorders [3] and muscular dysmorphia [4].” This is a simplistic view. I recommend the authors to expand this statement by mentioning the biological and psycho-social explanation.

In lines 35 and 36 we add:

“The relationship between BD and adviser mental health could be due to shame and social isolation [5] and hormonal changes such as leptin [6].”

2.     Under Introduction line 36, the authors stated “Similarly, BI is a key element of the self-concept configuration, especially in adolescence [5,6], when everything related to the body becomes especially relevant.” I think the authors need to be more specific because the statement “when everything related to the body becomes especially relevant is rather non-specific.

In lines 40 and 41 we add:

“Older age, male gender and lower body mass index were associated with better body esteem in adolescents [10].”

3.     Under discussion, line 292, the authors mentioned that “On the part of adiposity, a higher BMI was associated with higher BD [23]” I suggest the authors to indicate the relationship between high BMI and depression in female adolescents.

In line 312 was added:

“… and depression in adolescent females [59]”.

 4.     Line 373, under limitations, the authors should state the limitation of cross-sectional study and cannot establish causality.

This information was added in the limitations section (lines 396-397):

“This is a cross-sectional study and therefore causal relationships cannot be established.”

Reviewer 2 Report

This research aimed at explaining how sex, BMI and other associated variables to physical activity impact on adolescents' body dissatisfaction.

Introduction.

The introduction section is well argumented and presents variables and their associations with respect to study aims. Nontheless I think it would be important to define clearly the adolescence period, since this developmental stage could be divided in different age bands (pre-adolescence, Authors inquiry only some specific age range and should clarify the reasons of their decision.

Authors demonstrate that male and female differ with respect to body desidered image compared to their actual BMI highlighting gender specific relationships.  

Method

Information about participant recruitment should be given (were they be contacted via mail? Through the school districts?). Data on response rate should be added.

Table 1 reports data on descriptive statistics, but the title does not mention how statistical differences have been calculated (t test?). This information should be added in the data analysis section.

Information about factorial three way ANOVA should be given: a) authors tested the presence of outliers?; b) normality distribution of scores has been tested? c) homogeneity of variance was met?

MANCOVA: assumption for multicollinearity was tested? Partial eta squares should be reported.

Other minor concerns:

Line 50: [11]may; include space Line 44: body image and body satisfaction have been investigated also in at risk population as reported by adolescents healed form pediatric leukemia that was mostly below the 50 percentile (50%) comparing with norms, especially those that underwent a high risk leukemia (Tremolada et al., 2017). Please refer to  Tremolada M., Bonichini S., Taverna L., Basso G., Pillon M. (2017). Self-esteem and academic difficulties in preadolescents and adolescents healed from paediatric leukaemia, Cancers, 9(6): 55; doi:10.3390/cancers9060055 Line 66: gender better than sex Line 100: studying the interaction between gender and type of sport Lines 111-112: "expecially among males". I think this sentence should be better explained. Is it justifiedonly by the disproportionality of studies or authors are convinced that also males in recent years practice sports to lose weight? Authors should clear from the beginning of the article how they define adolescence, do they distinguish adolescence in different periods? Are there differences between males and females withrespect of age? Lines 121-126: More details should be given with respect to procedure recruitment and response rate. Line 129: The BSQ was adapted to the Spanish population. Authors should explain how the cultural adaptation was carried out. Line 167: have been tested Lines 300-301: I would use gender-specific, instead of sex-specific Line 303: gender instead of sex. Lines 310-311: other studies with adult samples Line 326: after citation a point and a space before capital letter Line 363: a point to end the sentence Line 370: from those Line 380: gender Line 389: gender specific

I wish to the authors success with their work.

Author Response

We would like to thank the reviewer for providing us with such a detailed assessment on our article, as well as for the comments reported to improve this work. In the following, we provide details on the enhanced aspects based on the reviewer’s suggestions.

REVIEWER´S COMMENTS

AUTHOR´S RESPONSE

CHANGES IN TEXT

Introduction

1. The introduction section is well argumented and presents variables and their associations with respect to study aims. Nontheless I think it would be important to define clearly the adolescence period, since this developmental stage could be divided in different age bands (pre-adolescence, Authors inquiry only some specific age range and should clarify the reasons of their decision.

Thank you for your considerations.

In page 1 line 38-40, we define “adolescence”. This definition is compatible with the age range chosen by the authors and is justified by two issues:

a)      WHO (2006) defines “adolescence” as the period of life beginning with the appearance of secondary sex characteristics and terminating with the achievement of full maturity (usually 12–18 years of age). It is indeed during this period when the main body image issues are initiated and developed (in lines 38-40).

b)     In the Spanish education system, the age range from 12 to 17 years corresponds with the secondary education grades which are compulsory (in method, line 131-133). This educational stage is carried out entirely in Secondary Schools.

Method

2. Information about participant recruitment should be given (were they be contacted via mail? Through the school districts?). Data on response rate should be added.

Thank you for this comment.  It is true that the information initially reported by the authors was not complete enough.

-         Data on procedures for contacting with schools and parents have been included (line 203-205).

-         Participants were recruited from the entire population of secondary education students in La Roda (Spain) (line 201). That is, all the secondary students from that place were invited. This is a town of approximately 13,000 inhabitants, with two secondary schools. The adolescent population is approximately 820 people.

-         Data on response rate added in line 137.

3. Table 1 reports data on descriptive statistics, but the title does not mention how statistical differences have been calculated (t test?). This information should be added in the data analysis section.

Thank you for this comment.

P-values were extracted from independent t-tests comparing male and female participants on the different included variables. A little clarification has been added in table 1 (line 258) and also in the data analysis section (lines 228 and 229).

4. Information about factorial three way ANOVA should be given:

a) authors tested the presence of outliers?;

b) normality distribution of scores has been tested?

c) homogeneity of variance was met?

In response to the questions:

a) The presence of outliers was tests. Those outliers based on mistakes during the data input were checked with the original questionnaires. Some of them were corrected and those with erroneous data were deleted. In summary, 18 participants were deleted, including cases considered outliers (line 136). On the other hand, z-values higher than 3 with no mistakes in data gathering or imputation were noticed: 6 outliers in BMI, 8 in BSQ scores and 4 in DDPB scores. These cases were not deleted as they were within the respective ranges of prevalence for the different variables and because their presence did not significantly influence the final results.

All this information was not included into the initial version of the article to not increase its extension. If the reviewer considers that it is convenient to include it, this request will be met.

b and c) Normality and homoscedasticity  were tested. Although these assumptions were met in several variables, they were not in others.  Notwithstanding, we decided to use parametric tests. Since the number of participants in this sample was high, normality could be assumed according to the central limit theorem. This has been included on page 5 lines 224-227.

5. MANCOVA: assumption for multicollinearity was tested? Partial eta squares should be reported.

Thank you for this suggestion.

-         Partial eta squares have been added in tables 4 to 6. Also a clarification has been made on data analysis section (lines 239-240).

-         Multicolinearity between independent variables included into the MANCOVA test could not be tested as these variables are not scalar, but nominal.

Other minor concerns

6. Line 50: [11]may; include space

Corrected. Now is in line 56: “[16] may”.

7. Line 44: body image and body satisfaction have been investigated also in at risk population as reported by adolescents healed form pediatric leukemia that was mostly below the 50 percentile (50%) comparing with norms, especially those that underwent a high risk leukemia (Tremolada et al., 2017). Please refer to  Tremolada M., Bonichini S., Taverna L., Basso G., Pillon M. (2017). Self-esteem and academic difficulties in preadolescents and adolescents healed from paediatric leukaemia, Cancers, 9(6): 55; doi:10.3390/cancers9060055

Thank you for your comment.

On page2 lines 48-49, the wording has been modified and the suggested reference has been included.

8. Line 66: gender better than sex

The word “sex” has been replaced by “gender” (line 72).

9. Line 100: studying the interaction between gender and type of sport

The word “sex” has been replaced by “gender” (line 107).

10. Lines 111-112:

a) "expecially among males". I think this sentence should be better explained. Is it justified only by the disproportionality of studies or authors are convinced that also males in recent years practice sports to lose weight?

b) Authors should clear from the beginning of the article how they define adolescence, do they distinguish adolescence in different periods?

c) Are there differences between males and females with respect of age?

Thank you for the comments.

The authors may not be clear enough in their writing.

a) Indeed, the authors' writing was not clear. In the previous sentence it is commented that most of the studies have focused on studying the desires to lose weight; something that is more typical among females and not quite suitable for men. Therefore, the intention in the subsequent sentence was to highlight the need to assess dissatisfaction from another perspective more frequent in the case of men: the desire to lose weight. In this sense, the wording on page 3 lines 117-121 has been modified to clarify this idea.

b) This suggestion has been previously addressed in comment 2.

c) In table 1 age data by gender are reported with no differences between both males and females. In addition, no differences were found in body dissatisfaction between males and females by age. This result was not reported in the manuscript to not increase the extension of the article. Also, it did not influence the final results. If the reviewer considers that it is convenient to include it, this request will be met.

11. Lines 121-126: More details should be given with respect to procedure recruitment and response rate.

Thank you. This issue has already been addressed in the second suggestion.

12. Line 129: The BSQ was adapted to the Spanish population. Authors should explain how the cultural adaptation was carried out.

Thank you for your comment. The authors may have wrongly explained that this questionnaire had been previously adapted and validated to the Spanish context by other authors (Raich et al., 1999). The writing has been modified and improved to specify this fact (line 140 and 141).

13. Line 167: have been tested

Corrected in line 178

14. Lines 300-301: I would use gender-specific, instead of sex-specific

Corrected in line 320.

15. Line 303: gender instead of sex.

Corrected in line 322.

16. Lines 310-311: other studies with adult samples

Changed in line 329

17. Line 326: after citation a point and a space before capital letter

Corrected in line 345

18. Line 363: a point to end the sentence

Corrected in line 382.

19. Line 370: from those

Corrected in line 392.

20. Line 380: gender

Corrected in line 404.

21. Line 389: gender specific.

Corrected in line 413.

22. I wish to the authors success with their work.

Thank you for your wishes.

Reviewer 3 Report

The aim of the study is to investigate the relationships between gender, BMI, PA and body dissatisfaction. It should be noted that the authors focus on adolescent women and men as well as on various criteria of physical activity (type of sport, quantity and organization).Generally, physical activity is measured relatively one-dimensionally. Therefore, this approach offers the opportunity to understand which quantity and or quality of physical activity is important for the development of a healthy body image. Similarly, studies show that this may also be differentiated gender specific. Likewise, the BMI is considered as another moderating variable. An assumption is that the relationship in men is curvilinear, he is linear in women. The paper is well structured and written. The relevant studies are taken up and lead to a convincing and complementary research question. The research topic is within the scope of the paper. From my point of view some questions needs to be answered for a final approval of the paper. 

Major comments 

In total 625 (12 – 17 year) participant filled in a questionnaire in a cross-sectional design. Therefore, no statements about the causality / the direction of effect are possible. Both in theory and later in the discussion, this should be more evident. It is true that so far there are hardly any longitudinal studies, but nevertheless the few should be specially worked out. In the discussion this argument is present, but is not interpreted within the context of socialization and selection hypothesis. This should be added. The measures are appropriate. It should be stressed that these are instruments validated in Spanish, so that the findings are well comparable with findings from other countries (In the reference, please translate the Spanish title into English, APA-Standard). Superficially, the analysis seems appropriate. I see two core problems that make it difficult for me to evaluate the data conclusively. On the one hand, it is the sample size in the subsamples and. E.g. in table two, the n-numbers of the subsamples are reported, but only on the first dimension. However, since further subgroups are formed in the Anovas: Gender (2) x PA practice (4) * BMI (4) = 24 it is not clear how many people remain in the individual cells. If the number is too low, this can lead to distortions in the statistics. I would like to add another note to this, please put Table 2, in the sample description. On the other, the statistical key figures are inadequate (see APA standard). Among other things, no effect sizes are reported, but they are necessary to assess the relevance of the differences. Also, table 1, what was conducted t-test, F-test? Already in theory the authors address a possible curvilinearity. However, this idea is not reflected in the analyzes. Correlations do not seem to be appropriate here. Likewise, it would be useful to discuss the level of correlations, which tend to be rather small, especially in terms of physical activity.

I did not finally understand the different procedure for the ANOVA (table 4) and the MANCOVA (table 5) in terms of the chosen variables. What was the rational for choosing PA practice in the ANOVA and Type PA/Organization for the MANCOVER? Wouldn’t it make sense to look first at the correlation f the different PA-measures. It might die possible to aggregate these.

Discussion

The discussion is appropriate except for the two indications that I have already given (socialization vs. selection hypothesis, cross section versus longitudinal section). However, I miss the necessary statistical key figures in the results section for a final evaluation. Therefore, I will not go further here.

However, I find the final sentence a bit too broad. On the basis of these findings, I do not yet see any far-reaching implications for the practice. But I find the data too narrow and partly indifferent with a view to the findings of other studies.

Minor comments

P6 l244: BSDQ (-…) was significantly correlated (..) with a greater DDPB. What is meant by “greater”?  Weekly MVPA was only correlated with a “lower” dissatisfaction . I supposed the authors refer here to a positive resp. negative correlation. The corraltion with MVPA are rather small. Table 3: BSQ *MVPA female. 0 -0.90 is probably an error, I guess it is r=0.090 if p-value is reportet, the “*” is not necessary Please present stats of the post hoc tests Table 5 please check the degrees of freedoms, why BMI df=1?, why Type of PA df=2 Pa practice df = 2 Tabel 5 please name the Variable “Organization” and “Type of PA” always in the same way please present the full stats for the ANOVA / MANCOVA /chi2 and add effect size. See APA Standard please present reliability and validity scores for the used instruments

Author Response

We would like to thank the reviewer for providing us with such a detailed assessment on our article, as well as for the comments reported to improve this work. In the following, we provide details on the enhanced aspects based on the reviewer’s suggestions.

REVIEWER´S COMMENTS

AUTHOR´S RESPONSE

CHANGES IN TEXT

1. In total 625 (12 – 17 year) participant filled in a questionnaire in a cross-sectional design. Therefore, no statements about the causality / the direction of effect are possible. Both in theory and later in the discussion, this should be more evident. It is true that so far there are hardly any longitudinal studies, but nevertheless the few should be specially worked out.

Thank you for this suggestion. We remarked this kind of research included into the theoretical background (pages 1-2 lines 43-44) and the discussion (page 10 lines 382-385, and 396-397)

2. In the discussion this argument is present, but is not interpreted within the context of socialization and selection hypothesis. This should be added.

Thank you for the suggestion. A clarification has been added on page 10 lines 382-385.

3. The measures are appropriate. It should be stressed that these are instruments validated in Spanish, so that the findings are well comparable with findings from other countries (In the reference, please translate the Spanish title into English, APA-Standard).

Thank you for your suggestions.

The Spanish validation of the instruments has been stressed. This can be seen on page 3 line 140.

Also, references’ titles have been properly translated on page 13 to 14 (references 48, 51, 54, 55 and 71).

4. Superficially, the analysis seems appropriate. I see two core problems that make it difficult for me to evaluate the data conclusively. On the one hand, it is the sample size in the subsamples and. E.g. in table two, the n-numbers of the subsamples are reported, but only on the first dimension. However, since further subgroups are formed in the Anovas: Gender (2) x PA practice (4) * BMI (4) = 24 it is not clear how many people remain in the individual cells. If the number is too low, this can lead to distortions in the statistics.

The reviewer makes indeed an accurate comment. Nevertheless, the aim of this research was to determine the relationship between BI and some primary variables (sex, BMI, type of PA, organisation of PA, etc.). This is why information on their respective frequencies is reported in table 2. Interactions among them could result into a high number of subgroups and differences among these, which would go further the scope of the present study. Also, in case of studying them, a much greater number of participant would be required. Despite this fact, since their specification is needed for the respective ANOVA and MANCOVA analyses, their combined effect is included in tables 4 and 5.

Notwithstanding, if the reviewer considered it appropriate, these data could be added.

5. I would like to add another note to this, please put Table 2, in the sample description

Thank you for your suggestion. Changes are displayed on page 6 lines 250 to 255.

6. On the other, the statistical key figures are inadequate (see APA standard).

We are sure that there could be some error on statistical key figures, but we are not able to notice it. In this line, however, IJERPH does not mention APA normative for its tables.

If the reviewer refers to the introduction of a “0” before "." in decimal numbers, we did it following the recommendations of this journal.

7. Among other things, no effect sizes are reported, but they are necessary to assess the relevance of the differences.

Eta squared stats have been added in table 1; and partial eta squared stats have been added in tables 4 to 6. Also a clarification has been made on data analysis section (lines 239-240).

8. Also, table 1, what was conducted t-test, F-test?

Thank you for this comment. p-values were extracted from independent t-tests comparing male and female participants on the different included variables. A little clarification has been added in table 1 (page 6 line 258-259) and also in the data analysis section (page 5, lines 229 and 230).

9. Already in theory the authors address a possible curvilinearity. However, this idea is not reflected in the analyzes. Correlations do not seem to be appropriate here.

Thank you for your comment.

The possible curvilinearity is suggested only for males from the results of DDPB for the different categories of BMI (see table 1), but not from the correlational analysis. This kind of evidence could not be extracted from the latter as it is a linear analysis.

We opted not to run complementary analyses because this may be beyond the main aims of this research and the article would become too long.

10. Likewise, it would be useful to discuss the level of correlations, which tend to be rather small, especially in terms of physical activity.

Thank you for your suggestion.

In the discussion (line 322) the small correlation is specified.

11. I did not finally understand the different procedure for the ANOVA (table 4) and the MANCOVA (table 5) in terms of the chosen variables. What was the rational for choosing PA practice in the ANOVA and Type PA/Organization for the MANCOVER? Wouldn’t it make sense to look first at the correlation f the different PA-measures. It might die possible to aggregate these.

Thank you for this comment. The wording might be confusing.

In the ANOVA test (table 4) the effect of the different variables (sex, BMI and PA practice) on BD was analysed. This analysis was run considering the whole simple including both active and inactive participants.

Once the importance of the BMI’s effect, but not PA practice’s effect, was confirmed, a MANCOVA test (table 5) is run (only with active participants); by controlling for the effect of BMI in order to determine whether there are or not differences according to the type of PA among active participants.

The type of PA could not be included into the ANOVA test (table 4), since not all the participants were sport practitioners.

12. Discussion

The discussion is appropriate except for the two indications that I have already given (socialization vs. selection hypothesis, cross section versus longitudinal section). However, I miss the necessary statistical key figures in the results section for a final evaluation. Therefore, I will not go further here.

However, I find the final sentence a bit too broad. On the basis of these findings, I do not yet see any far-reaching implications for the practice. But I find the data too narrow and partly indifferent with a view to the findings of other studies.

Thank you for your comments. We hope that the changes made had met the two indications stated.

Additionally, we have modified the last two sentences in order to not be too broad.

13. P6 l244: BSDQ (-…) was significantly correlated (..) with a greater DDPB. What is meant by “greater”?  Weekly MVPA was only correlated with a “lower” dissatisfaction . I supposed the authors refer here to a positive resp. negative correlation.

Perhaps the wording has not been good enough to make this result clear. The wording has been changed (line 264).

The intended meaning is that higher scores in the BSQ correlated with more discrepancy between the perceived and desired body. That is, participants wanted to be thinner than they were perceived by themselves. Therefore, the cognitive component of dissatisfaction correlates with the perceptive component.

On the other hand, as explained in the method, we would like to clarify that higher scores in the BSQ mean greater dissatisfaction; while in the case of the DDPB, higher scores mean a desire to gain weight and negative scores mean a desire to lose weight. Therefore, the DDPB correlates negatively with the BSQ and positively with the weekly MVPA.

14. The correlation with MVPA are rather small. Table 3: BSQ *MVPA female. 0 -0.90 is probably an error, I guess it is r=0.090 if p-value is reportet, the “*” is not necessary

Thank you for the correction. The mistake has been corrected in table 3.

15. Please present stats of the post hoc tests

The stats can be check in table 6 (Pairwise comparisons). If complementary information is considered necessary, we are willing to provide with it.

16. Table 5 please check the degrees of freedoms, why BMI df=1?, why Type of PA df=2 Pa practice df = 2.

BMI is a covariate; thus it is not included as another factor with subgroups.

Type of PA has 2 df which correspond to the 3 groups (non-aesthetic/non-lean—non-aesthetic/lean—and aesthetic/lean).

PA practice has 2 df which correspond to the 3 groups (inactive; actives <60 min/day of MVPA; and actives ≥60 min/day of MVPA).

17. Table 5 please name the Variable “Organization” and “Type of PA” always in the same way

Thank you for the correction. The mistakes have been corrected in table 5.

18. See APA Standard please present reliability and validity scores for the used instruments.

The internal consistency of BSQ for this research is presented in line 148.

Regarding the rest of the questionnaires, despite being validated and used in different studies in the Spanish context, they offer problems to present validity or reliability indices for this study.

In this sense, given the characteristics of Gardner’s scale (selection of pictures), reliability indexes cannot be presented since these could only be offered through test-retest tests. We consider that these tests are not necessary for this type of study, but for studies aiming to validate the questionnaire.

Finally, regarding the IPAQ-SF and PAQ-A questionnaires, these are widely used questionnaires. Despite this, it becomes difficult to offer validity and reliability data since these were not used in their entirety.

Round 2

Reviewer 3 Report

The authors have dealt comprehensively with the comments from the review. This has led to the clarification of my question. For example, from my point of view, the addition of effect sizes allows to better assess the relevance of the differences, which are indeed moderate to strong. Thus, the contribution has gained significance. The additions to the introduction and discussion have also strengthened the paper. Without going into further detail, I would like to sum up, that the article is a valuable addition to existing research and I can recommend a publication.